# Vapor Bubble Deformation and Collapse near Free Surface

Yue Chen [1], Qichao Wang [1], Hongbing Xiong [1,*] and Lijuan Qian [2,*]

1   State Key Laboratory of Fluid Power and Mechatronic Systems, Zhejiang University,
    Hangzhou 310027, China; cheny9924@163.com (Y.C.); 21824016@zju.edu.cn (Q.W.)
2   College of Mechanical and Electrical Engineering, China Jiliang University, Hangzhou 310018, China
*   Correspondence: hbxiong@zju.edu.cn (H.X.); qianlj@cjlu.edu.cn (L.Q.)

**Abstract:** Vapor bubbles are widely concerned in many industrial applications. The deformation and collapse of a vapor bubble near a free surface after being heated and raised from the bottom wall are investigated in this paper. On the basis of smoothed particle hydrodynamics (SPH) and the van der Waals (VDW) equation of state, a numerical model of fluid dynamics and phase change was developed. The effects of fluid dynamics were considered, and the phase change of evaporation and condensation between liquid and vapor were discussed. Quantitative and qualitative comparisons between our numerical model and the experimental results were made. After verification, the numerical simulation of bubbles with the effects of the shear viscosity $\eta_s$ and the heating distance $L$ were taken into account. The regularity of the effect of the local Reynolds number (Re) and the Ohnesorge number (Oh) on the deformation of vapor bubbles is summarized through a further analysis of several cases, which can be summarized into four major patterns as follows: umbrella, semi-crescent, spheroid, and jet. The results show that the Re number has a great influence on the bubble deformation of near-wall bubbles. For $Re > 1.5 \times 10^2$ and $Oh < 3 \times 10^{-4}$, the shape of the bubble is umbrella; for $Re < 5 \times 10^0$ and $Oh > 10^{-3}$, the bubble is spheroidal; and for $5 \times 10^0 < Re < 1.5 \times 10^2$, $3 \times 10^{-4} < Oh < 10^{-3}$, the bubble is semi-crescent. For liquid-surface bubbles, the Re number effect is small, and when $Oh > 5 \times 10^{-3}$, the shape of the bubble is jet all the time; there is no obvious difference in the bubble deformation, but the jet state is more obvious as the Re decreases. Finally, the dynamic and energy mechanisms behind each mode are discussed. The bubble diameter, bubble symmetry coefficient, and rising velocity were analyzed during their whole processes of bubble growth and collapse.

**Keywords:** vapor bubble; deformation; SPH method; phase change; Re number





## 1. Introduction

A vapor bubble is a kind of gas that is generated by the instantaneous injection of high energy to liquid, such as laser, electricity, or other rapid heating methods. It has been applied in many industries; for example, the heat transfer of two-phase heat exchangers, surface corrosion caused by shock waves, injection without needles, and destruction of biological tissue using liquid jet superheat transfer [1–4]. In recent years, the study of vapor bubbles has attracted a lot of attention, but the dynamic process of vapor bubbles is nonlinear and highly complex [5,6]. It includes bubble oscillation and interface fluctuation during bubble growth, shock wave impact, and cavitation noise during bubble collapse. Understanding the deformation and collapse mechanisms of vapor bubbles is the key to successfully address these application-related challenges.

In order to study the dynamics of vapor bubbles, many researchers were involved in experimental research and numerical simulations. In experiments, the usual way to create vapor bubbles is to use pulsed lasers or electric sparks to shoot them instantaneously, and to observe the interaction of the bubbles with the solid surface using high-speed cameras [7,8]. Gonzalez et al. [9] studied the dynamic process of laser-induced bubbles in liquid gaps at different heights. Sun et al. [10,11] concluded that the thermal effect plays an important role

in the growth and collapse of vapor bubbles in microchannels. Kangude et al. [12] studied the cavitation mechanism of vapor bubbles on hydrophobic surfaces using the thermal imaging method. The experimental results of Tang showed that the hydrophobic coating has a significant influence on the growth process of laser-induced bubbles [13]. According to the three-dimensional view of the experiment, Hement et al. [14] found that the near-wall bubble collapses very rapidly, and that the tangential flow would lead to the formation of the ring cavity. Considering the influence of gravity and viscosity, Sangeeth et al. [15] studied the jet velocity resulting from bubble collapse at a liquid surface; the dependence of the dimensionless jet velocity, expressed in terms of the Weber number, on the Bond number, is determined by the dimensionless cavity depth. The Weber number is used to measure the relationship between the surface tension and the inertia force, and the Bond number is used to measure the relationship between the gravity and the surface tension. By using a spark-generated device and a high-speed camera, Zhang et al. [16] conducted an experiment on the dynamic process of vapor bubbles under water; six classical water burial phenomena were induced, and their forming mechanism was analyzed. Ma et al. [17] investigated the growth of vapor bubbles under different levels of gravity by conducting an experiment and found that gravity has a significant influence on the growth of a single vapor bubble.

Besides experiments, many numerical methods have also been reported to analyze the bubble dynamics, including the Lattice Boltzmann method (LBM), the finite difference method (FDM), the finite volume method (FVM), the volume of fluid method (VOF), the smoothed particle hydrodynamics method (SPH), and so on. Liu et al. [18] deduced the relationship between the various characteristic parameters of bubbles, then explored that different liquid parameters would have a significant impact on the cavitation process based on the FDM. Phan et al. [19] used a compressible homogeneous mixture model to numerically investigate the dynamics of an underwater explosion bubble, and the dynamic bubble motion including the bubble expansion, contraction, collapse, jet, and rebound. As for the VOF methods, they contain the algebraic VOF and geometric VOF methods. Owing to the merits of mass conservation, the latter was widely applied in the bubble simulation in [20]. By using the VOF method, Tang [13] simulated the hydrophobic wall surface by controlling the thickness of the air film at the solid–liquid interface, studied the oscillation behavior of the laser-induced bubble, and summarized the dynamic mechanism and law. Nguyen et al. [21] used a geometrical VOF algorithm based on the piecewise–linear interface calculation (PLIC) to numerically investigate the dynamic behavior of bubble collapses, water jets, and pressure loads during the collapse of the bubble near walls and a free surface; the results showed a good agreement between the simulation and experiment of the bubble dynamics during the collapse process. Erin et al. [22] used the SPH and VOF methods to simulate the rising of bubbles, and compared with previous experimental results, they concluded that both the VOF and SPH methods may be used to capture physically realistic transient and steady-state multi-phase systems; the SPH method could better capture the centroid of the bubble, while the VOF method better captured the rising velocity of the bubble.

SPH is a meshless method. It is uniquely capable of representing the dynamic evolution of complicated geometries without additional algorithmic complication, such as those found in multi-phase flows [22]. When simulating fluid dynamics problems, the SPH method discrete the flow field into moving fluid micro clusters, which can be regarded as a combination of a series of molecules with the same properties [23]. Unlike the traditional grid algorithm, SPH has no grid connection between the particles, and follows the interaction between the particles, which is suitable for any large deformation problem. On the other hand, the SPH method uses the Lagrange method to describe the flow field, which can be used to study some multiphase flow problems with discrete phases and has obvious advantages over the traditional grid algorithm in the study of large deformation and dynamic boundary problems.

The diffusion interface method (DIM), based on the SPH method, is more commonly used when involved in the precise capture of the gas–liquid phase flow interface [24]. Sigalotti et al. [25] first used the DIM, which treats the gas–liquid interface as continuous, added the Korteweg tensor to characterize the capillary forces, and used the SPH algorithm [23] to solve, which was proven to be useful in cavitation hydrodynamics. Gallo et al. [26] verified the study on the nucleation of vapor bubbles in metastable liquid by using the DIM. Wang [24] applied the DIM to numerically simulate the rising of vapor bubbles in static water, and proved that it is feasible to calculate the dynamics of vapor bubbles by using the DIM. Moreover, the relation between the shape of the rising vapor bubble and the dimensionless parameters such as the Reynolds number was also introduced.

Based on the gas–liquid DIM, the Navier–Stokes–Korteweg (NSK) equation considering the gas–liquid interfacial tension is derived, the van der Waals (VDW) equation of state is introduced, and the SPH algorithm is used for the numerical solution. The effects of the shear viscosity $\eta_s$ and the heating distance $L$ on the growth and collapse processes of the vapor bubble are taken into account. The regularity of the effect of the Re number and the Oh number on the deformation of vapor bubbles is summarized through a further analysis of several cases, which can be summarized into four major patterns. Then, the formation mechanism is analyzed, and the growth and collapse of the bubbles are studied. According to our results, it is possible to precisely control the deformation of vapor bubbles by adjusting the two dimensionless parameters, the Re number and the Oh number. This has a certain engineering guiding significance for the current application, for example, the avoidance of surface corrosion caused by nuclear boiling, the dispersion of poisonous droplets caused by the disintegration of vapor bubbles, and the effect of bubble deformation on EHD-enhanced boiling heat transfer [27].

## 2. SPH Modeling

In our model, compressible vapor and liquid are considered to be two-phase fluids with a continuous density gradient. In the Lagrange formula, the liquid and gas phases uniformly follow the conservation equations of mass, momentum, and energy as follows:

$$\frac{d\rho}{dt} = -\rho \nabla \cdot \mathbf{v} \tag{1}$$

$$\rho \frac{d\mathbf{v}}{dt} = \nabla \cdot \mathbf{M} + F_E \tag{2}$$

$$\frac{d\mathrm{U}}{dt} = \frac{1}{\rho}\mathbf{M} : \nabla\mathbf{v} + \frac{\kappa}{\rho}\nabla^2 T \tag{3}$$

where $\rho$ is the density, $\mathbf{v}$ is the velocity vector, $\mathbf{M}$ is the stress tensor, $F_E$ is the external force of gravity, $T$ is the temperature, U is the internal energy, and $\kappa$ is the thermal conductivity. The stress tensor $\mathbf{M}$ includes the pressure terms, the shear and bulk viscosity terms, as well as an additional Korteweg tensor $\mathbf{M}_c$ of the gas–liquid diffusion interface, as follows:

$$\mathbf{M} = -p\mathbf{I} + \eta_s(\nabla\mathbf{v} + \nabla\mathbf{v}^T) + (\eta_v - \frac{2}{\dim}\eta_s)(\nabla \cdot \mathbf{v})\mathbf{I} + \mathbf{M}_C \tag{4}$$

where $p$ represents the pressure, dim represents the dimension of space, and $\eta_s$ and $\eta_v$ are the shear and volume dynamic viscosity, respectively. The Korteweg tensor $\mathbf{M}_c$ can be used to simulate the capillary force on the interface due to the density gradient, expressed as follows:

$$\mathbf{M}_C = K(\rho\nabla^2\rho + \frac{1}{2}|\nabla\rho|^2)\mathbf{I} - K\nabla\rho\nabla\rho \tag{5}$$

where $K$ is the gradient energy coefficient for a given material.

According to the description of the SPH model, the density can be better described in the following summation density Equation (6), instead of the continuity Equation (1).

The summation density conserves mass exactly and guarantees second-order accuracy [25], which benefits the simulation of the liquid–vapor interface.

$$\rho_a = \sum_b \frac{m_b}{\rho_b} \rho_b W_{ab} = \sum_b m_b W_{ab} \tag{6}$$

where $m$ is the particle mass, the subscript $b$ indicates the neighbor particles around this particle $a$, the subscript $ab$ denotes the variable difference between particle $a$ and $b$, the subscript $b$ represents the adjacent particles around particle $a$, and $W_{ab}$ is a kernel function, which explains the particle distance between particles $a$ and $b$.

The momentum and energy calculations are discretized into long-range and short-range terms, because the same smoothing length for all the force terms was unable to handle the surface tension effects and cause the interfacial instability [28] as follows:

$$\frac{dv_a}{dt} = \sum_b m_b \left( \frac{\mathbf{M}_a}{\rho_a^2} + \frac{\mathbf{M}_b}{\rho_b^2} \right) \cdot \nabla W_{ab} + \sum_b m_b \left( \frac{\mathbf{M}_a^H}{\rho_a^2} + \frac{\mathbf{M}_b^H}{\rho_b^2} \right) \cdot \nabla W_{ab}^H + F_E \tag{7}$$

$$\frac{dU_a}{dt} = \frac{1}{2} \sum_a m_a \left( \frac{\mathbf{M}_a}{\rho_a^2} + \frac{\mathbf{M}_b}{\rho_b^2} \right) : v_{ba} \nabla W_{ab} + \frac{1}{2} \sum_b m_b \left( \frac{\mathbf{M}_a^H}{\rho_a^2} + \frac{\mathbf{M}_b^H}{\rho_b^2} \right) : v_{ba}^H \nabla W_{ab}^H + U_E \tag{8}$$

where the short-range repulsive term has the smooth length of $h$ with no mark, and the double smooth length of $H = 2h$ is used for the long-range attractive term and marked with superscript $H$. $\mathbf{M}$ and $\mathbf{M^H}$ are equal to $\mathbf{M} - \bar{a}\rho^2\mathbf{I} - \mathbf{M}_C$ and $\mathbf{M} + \bar{a}\rho^2\mathbf{I}$, respectively.

In this paper, we use the hyperbolic kernel function proposed by Yang [29], which ensures that the distribution of particles is more uniform in both the two-dimensional and three-dimensional problems and does not lead to the unstable growth of stress, so better results can be obtained in the simulation.

In order to close the momentum and energy equations, the VDW equation is chosen to describe the pressure state equation, which can describe the gas–liquid coexistence system. The expression of the van der Waals equation of state is as follows:

$$p = \frac{\rho \bar{k}_b T}{1 - \bar{\beta}\rho} - \bar{\alpha}\rho^2 \tag{9}$$

where $\bar{k}_b$ is the Boltzmann constant, $\bar{\alpha}$ is the parameter of attraction, and $\bar{\beta}$ is related to the size of the particles. And the critical state is expressed by these three parameters:

$$T_c = \frac{8\bar{\alpha}}{27\bar{k}_b\bar{\beta}}, \; \rho_c = \frac{1}{3\bar{\beta}}, \; P_c = \frac{\bar{\alpha}}{27\bar{\beta}^2} \tag{10}$$

where $\bar{k}_b$, $\bar{\alpha}$, and $\bar{\beta}$ are set as 1, 2, and 0.5 for the VDW fluid, respectively. Here, the gas or liquid phase is distinguished by the critical density of the VDW fluid. According to the VDW isothermal curve [30], when $\rho \to 0$, the VDW equation transforms into the ideal gas law. Therefore, if the fluid density is less than the critical density, it is the gas phase. Otherwise, it is the liquid phase.

Using the SPH discretization Equations (6) to (8) and coupled with the VDW EOS Equation (9), the liquid and the heated vapor are simultaneously simulated. After that, the bubble position, velocity, size, and other properties are analyzed. More key parameters could be characterized by the following dimensionless numbers: Re represents the Reynolds number, which is used to measure the relationship between the inertia force and the viscous force, and Oh represents the Ohnesorge number, which is used to measure the relationship between the viscous force, the inertia force, and the surface tension. Their expressions are as follows:

$$\text{Re} = \frac{\rho \bar{v} D_{\max}}{\eta_s} \tag{11}$$

$$Oh = \frac{\eta_s}{\sqrt{\rho \sigma R_{\max}}} \qquad (12)$$

where $\rho$ represents the liquid density, $\bar{v}$ represents the average velocity of the bubble, $D_{max}$ represents the maximum diameter, $R_{max}$ represents the maximum radius as $R_{max} = D_{max}/2$, $\sigma$ represents the surface tension coefficient, and $\eta_s$ represents the dynamic viscosity coefficient.

The NSK equation is simulated in a non-dimensional scale; thus, all the data are presented without specified units in the following part. It is convenient to examine those bubble and liquid characteristics with the key non-dimensional parameters of the Re and Oh numbers. In the meantime, these non-dimensionalized parameters could be referred to dimensional ones using the given material properties. Here, for the water and water vapor bubble, the reference values in the length, temperature, time, and mass scales are $5.33 \times 10^{-8}$ m, 546 K, $1.36 \times 10^{-10}$ s, and $7.33 \times 10^{-20}$ kg, respectively [30].

## 3. Validation

### 3.1. Comparison Verification

In this section, we use the SPH method to simulate the water vapor bubble interaction with the free surface in two-dimension and compare it with the experimental results [16]. Our simulation setting is shown in Figure 1.

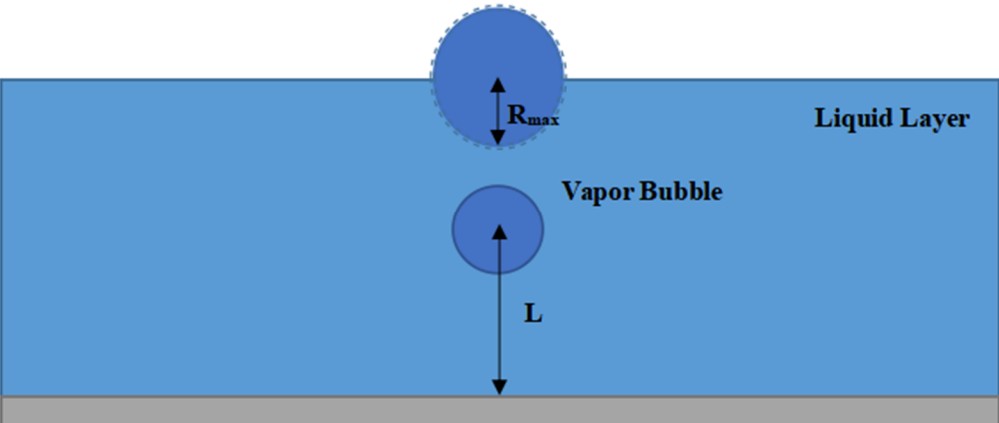

**Figure 1.** Vapor bubble heating geometry in our simulation.

Figure 2 shows the bubble deformation photos from Zhang's experiment with a bubble radius of about 15 mm. With the current computational efficiency and memory storage, it is still challenging to directly simulate the millimeter scale problem for our SPH simulation. Thus, we mimic the experimental setup in the following two critical parameters: the degree of super heat temperature $\Delta T$, the relative heating distance $\gamma_f$.

Firstly, we estimated the power input into the bubble and, thereafter, the bubble's super heat temperature. The experiment was conducted using a capacitor discharge with U = 200 V before the discharge, and 150 V after the discharge, with a capacity of 6600 μF and a thermal efficiency of 2%. Thus, the heat input to the bubble was about 1.155 J [16]. Using the saturated water vapor density of 0.6 kg/m$^3$ and a heat capacity of 2.075 J/g·K, we obtained the super heat temperature for a 15 mm water vapor bubble of about 65,622 K, which is non-dimensional as $\Delta T = 12$ in our system.

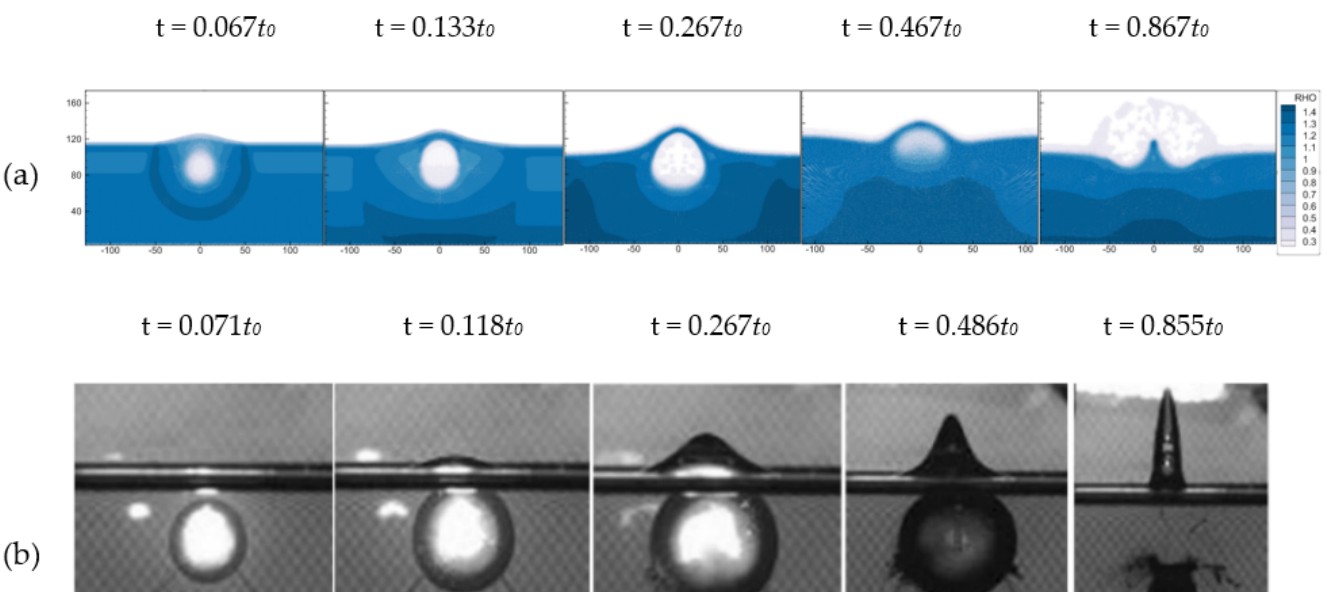

**Figure 2.** Comparison of the numerical results of SPH method and experiments. (**a**) Bubble deformation of $\gamma = 0.79$ in SPH simulation; (**b**) bubble deformation of $\gamma_f = 0.78$ in Zhang's experiment [16].

Then, we used the dimensionless distance between the burst point and the free liquid surface with the ratio to the bubble radius in Zhang's experiment as $\gamma_f = 0.78$ to arrange the heating position in our numerical simulation. We provide the data of two non-dimensional distances instead of one; $\varepsilon = H/R_{max}$ represents the dimensionless parameter of the liquid layer thickness, and $\lambda = L/R_{max}$ represents the dimensionless parameter of the bubble heating distance to the bottom of wall. Thus, the relative heating distance to the free surface could be obtained as $\gamma = \varepsilon$-$\lambda$, which has the same meaning as in Zhang's experiment of $\gamma_f$. Here, we used the height of the liquid layer $H = 160dx$ and the heating distance $L = 120dx$, where dx represents the initial distance between the SPH particles. After the simulation, we calculated $\varepsilon = H/R_{max} = 2.22$, $\lambda = L/R_{max} = 1.43$; therefore, $\gamma = \varepsilon - \lambda = 0.79$, which is approximately equal to the $\gamma_f$ value of 0.78 in Zhang's experiment.

We compare our simulation bubble deformation with the experimental photos of [16] during the bubble growth and collapse near the free surface. The lifetime of the bubble during its first cycle period is set as $t_0$ either for the experiment or simulation. Figure 2 shows the typical bubble deformation from its generation to its collapse in our SPH simulation and Zhang's experiment under a similar heating degree and geometry. Figure 3 shows the dimensionless radius $R/R_{max}$ during the first bubble cycle $t_0$, where $R$ represents the local bubble radius, and $R_{max}$ represents the maximum bubble radius. In the early period of the bubble growth, the bubble remains spherical; then, the bubble expands upwards along with the flow curve; finally, a clear upwelling column of water can be observed near the free surface. The results show that at the early stages before $0.3t_0$, our results are in good agreement with the experiments, either qualitatively or quantitatively. However, at the late stages after $0.3t_0$, there is a certain deviation between the numerical prediction and the experimental observation. For example, at $0.467t_0$, our simulation bubble has a more severe deformation, and a smaller jet height at $0.867t_0$, compared to the experiments. The reason is because the simulation accuracy is not that good for the SPH method; therefore, the computational error would be accumulated at the late stages. The advantages of SPH is the efficiency and flexibility in solving large deformation problems. In the future, we might strive to improve the accuracy of the SPH method. Currently, we believe that our SPH model is basically correct, and could be used to capture the bubble deformation during its deformation and collapse near the free surface.

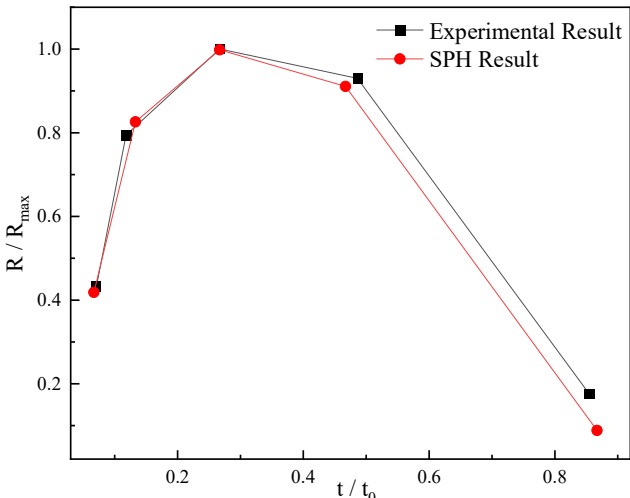

**Figure 3.** Comparison of SPH results with experiments for non-dimensional radius $R/R_{max}$.

*3.2. Shock Wave*

The second benchmark case is for the propagation of a shock wave. During the bubble's growth and collapse, the shock wave was found to be as important as the kinetic energy and thermal energy as shown in Figure 4. We tested the simulation of the discontinuity point in a typical one-dimensional shock wave problem using our SPH model.

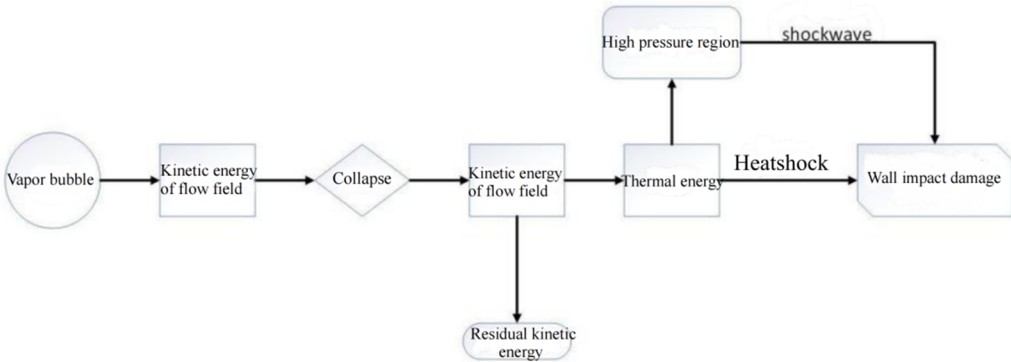

**Figure 4.** Energy transformation of vapor bubble.

Figure 5 shows the typical settings for a shock wave problem, and the discontinuity point is at $x = 2.5$. The initial state of the fluid on either side of the discontinuity point is $[\rho_L \ v_L \ p_L \ u_L] = [10 \ 0 \ 100 \ 25]$ and $[\rho_R \ v_R \ p_R \ u_R] = [1 \ 0 \ 1 \ 2.5]$. The specific heat capacity at constant volume is $c_v = 1$. The equation of state is as follows:

$$p = (\gamma - 1)\rho u = (\gamma - 1)c_v \rho T = 0.4 \cdot \rho T \tag{13}$$

where the subscripts $L$ and $R$ represent the left and right sides of the fluid. The total number of SPH particles is 1100, and the smooth length is $h = 1.8dx$.

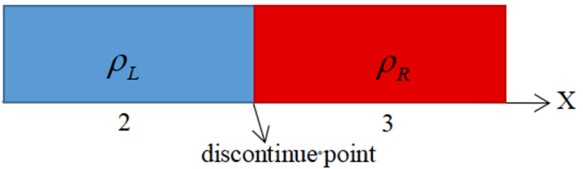

**Figure 5.** The settings of the shock wave problem.

The results for when the time $t = 0.4$ are shown in Figure 6. The red scatter is the result of the SPH, and the black line is the result of the exact Riemann solver. The SPH results agree well with the analytical solutions. This shows that our method is reliable in simulating the flow caused by the density and pressure difference.

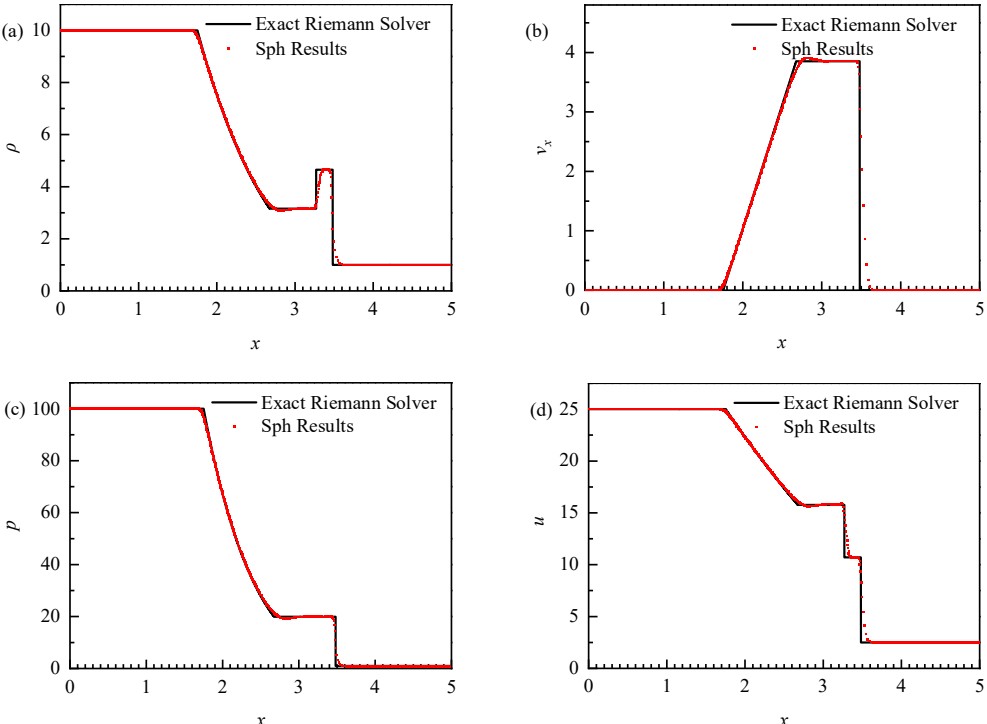

**Figure 6.** Comparison of the numerical results of the SPH method and the theoretical results of the exact Riemann solver. (**a**–**d**) show the distribution of density, velocity, pressure, and internal energy in the x direction, respectively.

## 4. Numerical Settings

The critical fluid density is introduced to distinguish between the liquid and vapor. The SPH liquid particle mass is $m = 0.6$, the stable density is $\rho = 1.2029$, the initial fluid temperature is $T_b = 1.01$, the heating height $L$ above the solid wall is set to be $L = 15\sim120$, the shear viscosity is $\eta_s = 0.1\sim1.0$ as shown in Table 1, the volume viscosity is $\eta_v = 0.5\eta_s$, the $400 \times 160$ particles are arranged at the bottom area in the x and y directions, the heating radius is $r = 12dx$, and the excess heat is $\Delta T = 12$. The region is uniformly spherical after laser heating. The left and right boundaries are periodic boundaries, and the upper and lower wall are set as adiabatic solid boundaries, which can be referred to in our previously published study [30]. The vapor bubble heating geometry in our simulation is shown in Figure 1. The values of the three dimensionless parameters $\gamma$, $\varepsilon$, and $\lambda$, defined in Section 3.1, are shown in Table 2.

**Table 1.** The parameter settings of each case.

| $\eta_s$ | CASE ID | L/dx |
|---|---|---|
| 1.0 | C1 | 15 |
| | C2 | 30 |
| | C3 | 60 |
| | C4 | 100 |
| | C5 | 120 |
| 0.1 | C6 | 15 |
| | C7 | 30 |
| | C8 | 60 |
| | C9 | 100 |
| | C10 | 120 |

**Table 2.** The dimensionless parameters of the bubble.

| CASE ID | H/dx | L/dx | $\varepsilon = H/R_{max}$ | $\lambda = L/R_{max}$ | $\gamma = \varepsilon - \lambda$ | $N_{secb}$ |
|---|---|---|---|---|---|---|
| C1 | 160 | 15 | 4.31 | 0.32 | 3.99 | 0 |
| C2 | 160 | 30 | 3.78 | 0.57 | 3.21 | 0 |
| C3 | 160 | 60 | 3.55 | 1.06 | 2.48 | 0 |
| C4 | 160 | 100 | 2.97 | 1.48 | 1.48 | 0 |
| C5 | 160 | 120 | 2.22 | 1.43 | 0.79 | 0 |
| C6 | 160 | 15 | 2.18 | 0.16 | 2.02 | 3 |
| C7 | 160 | 30 | 2.36 | 0.35 | 2.01 | 3 |
| C8 | 160 | 60 | 2.68 | 0.80 | 1.87 | 4 |
| C9 | 160 | 100 | 2.78 | 1.39 | 1.39 | 0 |
| C10 | 160 | 120 | 2.29 | 1.37 | 0.92 | 0 |

## 5. Results

### 5.1. Category of Bubble Deformation

First of all, the numerical simulations were performed for each case, and the effects of the shear viscosity $\eta_s$ and the heating distance $L$ on the growth and collapse processes of the vapor bubble are taken into account. The regularity of the effect of the Re number and the Oh number on the deformation of the vapor bubbles is obtained through a further analysis of several cases, which can be summarized into four major patterns. The following category between the Re and Oh numbers and the bubble deformation is drawn.

According to Figure 7, the liquid-surface vapor bubble takes on the jet shape, which is not greatly affected by the Re number. On the other hand, the near-wall vapor bubble varies in shape depending on the Re number and the Oh number. The shape can be divided into umbrella, semi-crescent, and spheroid. When $Re > 1.5 \times 10^2$ and $Oh < 3 \times 10^{-4}$, an umbrella shape is observed; when $Re < 5 \times 10^0$ and $Oh > 10^{-3}$, a spheroidal shape is present; and when $5 \times 10^0 < Re < 1.5 \times 10^2$, $3 \times 10^{-4} < Oh < 10^{-3}$, the bubble is categorized as semi-crescent.

Differences in the bubble deformation are notable between the liquid-surface bubbles (bubbles at high heating distance) and the near-wall bubbles (bubbles at medium and low heating distance). The high temperature in the vapor bubble of the near-wall bubble is absorbed by the solid wall and the thick liquid layer, leading to a less-pronounced deformation. Furthermore, the surface tension effect on the vapor bubble growth and collapse varies with the Re number and Oh number, indirectly influencing the limiting effect of the liquid, and ultimately leading to a differing bubble deformation.

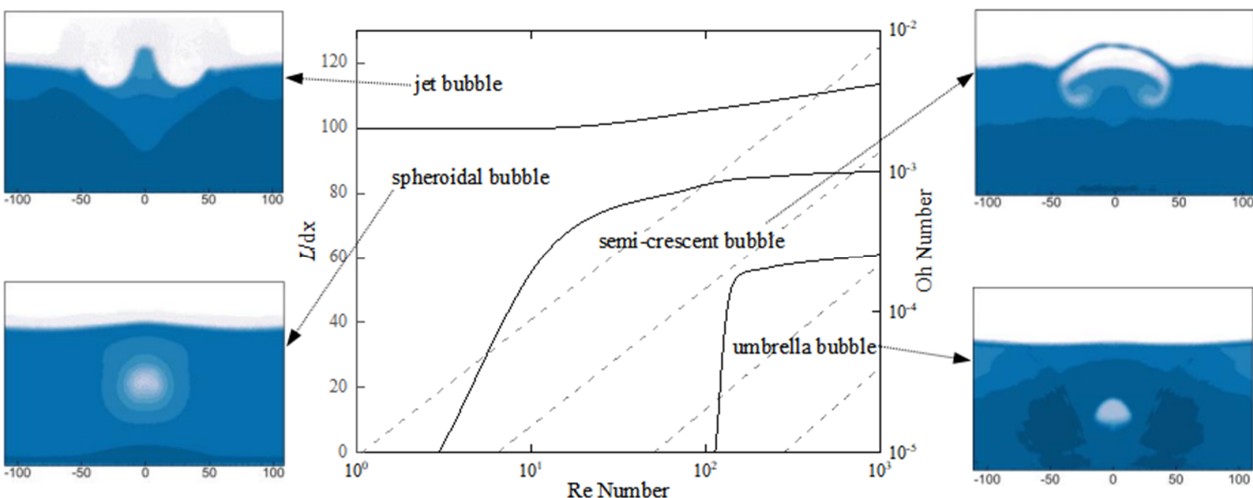

**Figure 7.** Category of bubble deformation of each mode.

### 5.2. Mechanism Discussion

#### 5.2.1. Jet Bubble

An analysis of the liquid-surface bubbles is carried out. Figure 8 depicts the formation of the jet bubble in case C5. With Oh $\approx 2 \times 10^{-3}$, during the initial growth phase, the bubble retains a spherical shape, and the free surface curves at $t = 50$. However, due to the existence of hydraulic fluid and the top of the bubble being drawn into the liquid surface, the bubble eventually takes on an oblong shape with a sharp top. Subsequently, the bubble gradually collapses and fuses with the liquid surface. This fusion produces a noticeable up-welling column of water at the point of contact. While capillary waves diminish in magnitude as the Oh number increases, the boundary of the bubble is smooth, and the impulse at its bottom increases, leading to an escalation in the jet velocity; this phenomenon is referred to as the jet bubble. The upper level of the liquid experiences significant oscillations caused by the rapid evaporation process, which affects the amount of heat energy present.

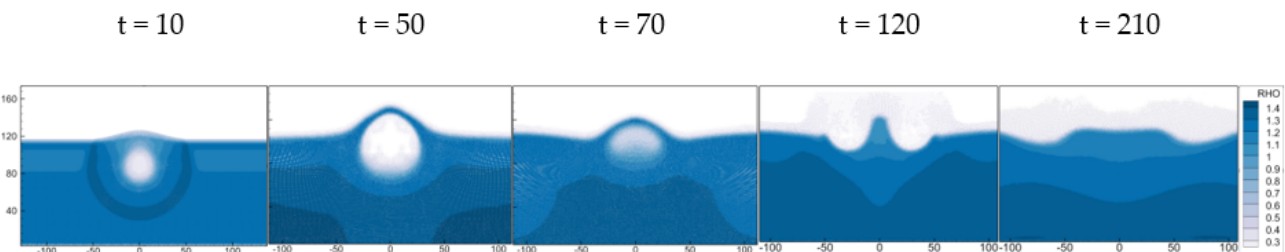

**Figure 8.** Deformation of jet bubble in case C5.

#### 5.2.2. Spheroidal Bubble

The near-wall bubble is investigated for its sensitivity to the different Re and Oh numbers. The spheroidal bubble shape change diagram in case C3 (Figure 9) demonstrates that the bubble maintains a spherical shape during the growth and collapse processes while Re $\approx 1.5$ and Oh $\approx 10^{-3}$, with the Re number approaching 1 indicating that the bubble's viscosity force is nearly equal to its inertia force, and the bubble maintains a spherical shape in the growth and collapse processes. The bubble reaches the minimum shape when $t = 120$, while the fluid zone phase is completed with minimal fluctuation in the liquid level. Due to the distance between the vapor bubble's center and the top liquid layer, the energy exchange can only occur within the liquid layer. A slight oscillation occurs at the top of the liquid at the end of the bubble collapse.

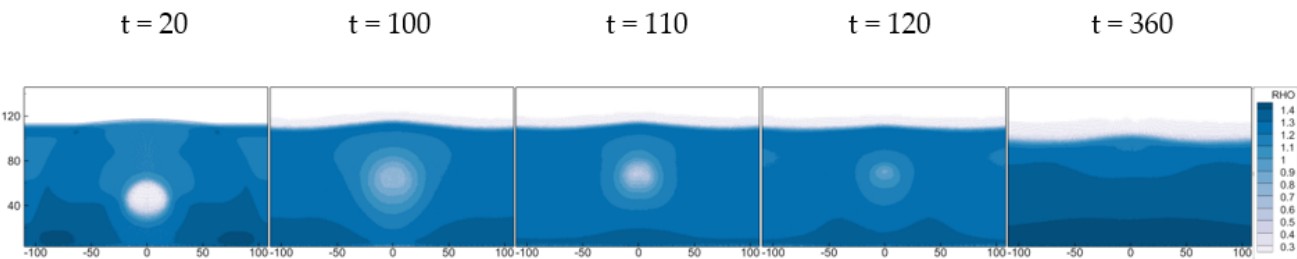

**Figure 9.** Deformation of spheroidal bubble in case C3.

### 5.2.3. Umbrella Bubble

For case C8, Figures 10 and 11 depict the density change and flow field vector diagrams of the umbrella bubble. The Re $\approx$ 150, and Oh $\approx 10^{-4}$. Initially, the bubble experiences extrusion pressure from both the left, right, and lower sides of the liquid, resulting in a longer flow field vector in the lower part of the bubble than in the upper part. As the Oh number reaches the minimum, due to the progressive damping of the capillary waves decreasing at $t = 50$, the bubble's edge becomes more irregular, causing it to transform from the ellipse to the umbrella shape. Throughout the process, buoyancy generates an upward force on the bubble, causing it to ascend. This results in the "umbrella handle" beneath the bubble to contract inwards, culminating in the formation of the fan bubble at $t = 70$.

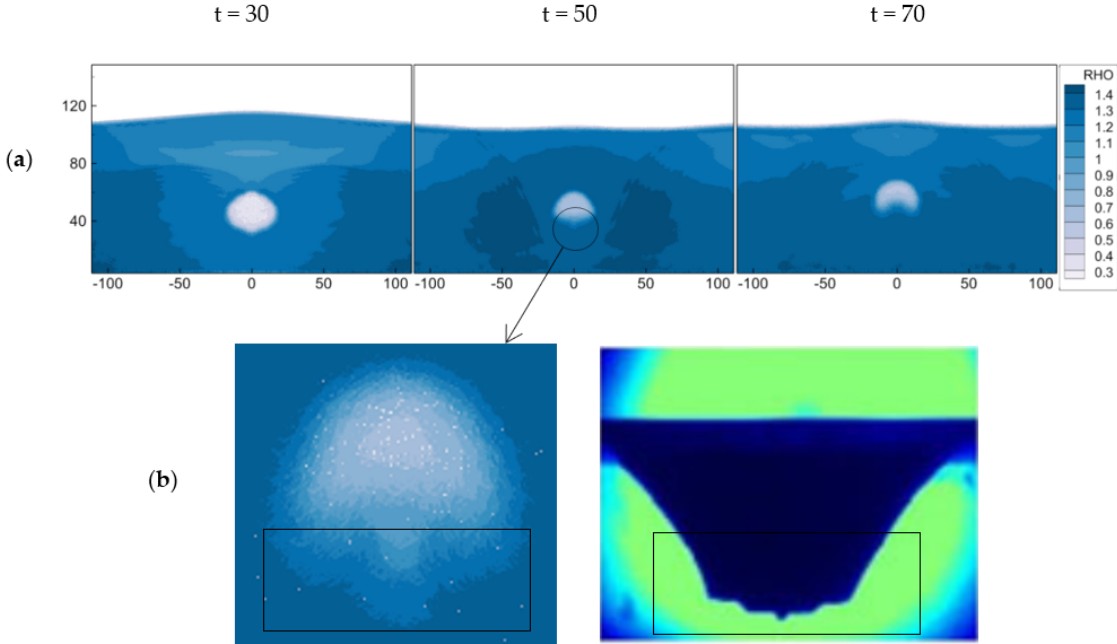

**Figure 10.** (**a**) Deformation of umbrella bubble in case C8; (**b**) capillary waves compared with Sangeeth's experiment [25].

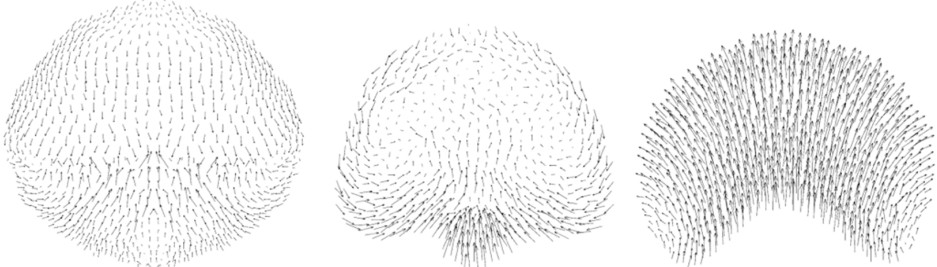

**Figure 11.** The flow field vector diagram of umbrella bubble in case C8.

### 5.2.4. Semi-Crescent Bubble

Figure 12 shows the density change and flow field vector diagram of the semi-crescent bubble in case C8. The Re $\approx$ 80, and Oh $\approx 8 \times 10^{-4}$. We found that at $t = 230$, the gas in the bubble moved up to the liquid surface to form a liquid film, the liquid film did not break during the expansion, and a thin "liquid bridge" appeared on the top of the liquid surface. Subsequently, when $t = 270$, the surface tension of the "liquid bridge" prevented the bubble from breaking, and it started to shrink downwards due to the force of the "liquid bridge". The flow field vector diagram in Figure 12b shows that due to the high temperature gas in the vapor bubble interacting with the free liquid surface, the "liquid bridge" vaporized, and small vapor bubbles formed quickly. Eventually, the vapor bubble collapsed on the free surface.

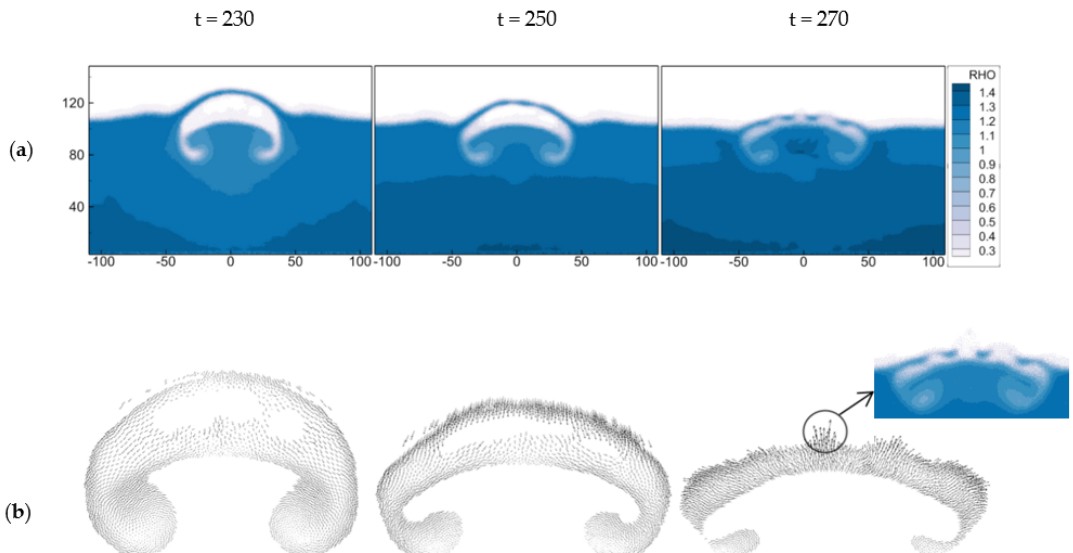

**Figure 12.** (**a**) Deformation of semi-crescent bubble; (**b**) the flow field vector diagram of semi-crescent bubble.

### 5.3. Bubble Deformation when Re < 10

The current analysis focuses on classical cases involving near-wall bubbles and liquid-surface bubbles when Re < 10, building on previous research. Figure 13 illustrates the variation in the bubble deformation and flow field density across cases C1~C5.

Figure 14 illustrates the variations in the longitudinal diameter of the near-wall bubbles and liquid-surface bubbles. The analysis indicates a marked difference in the deformation of the bubbles. For the near-wall bubbles (observed in cases C1~C3), the forces of viscosity and inertia are approximately equal, thereby enabling the bubbles to maintain their spherical shape. The bubbles undergo a growth stage, followed by a complete collapse throughout 3~4 cycles. For the liquid-surface bubbles (observed in cases C4~C5), the bubbles are subjected to an inertial force from below that causes them to make contact with the liquid surface and form the "liquid bridge"; then, by capillary wave damping, the impulse at the bottom of the bubbles increases, which eventually leads to the formation of the "jet". The two stages are discussed in detail below to allow for a better understanding of the bubbles' behavior.

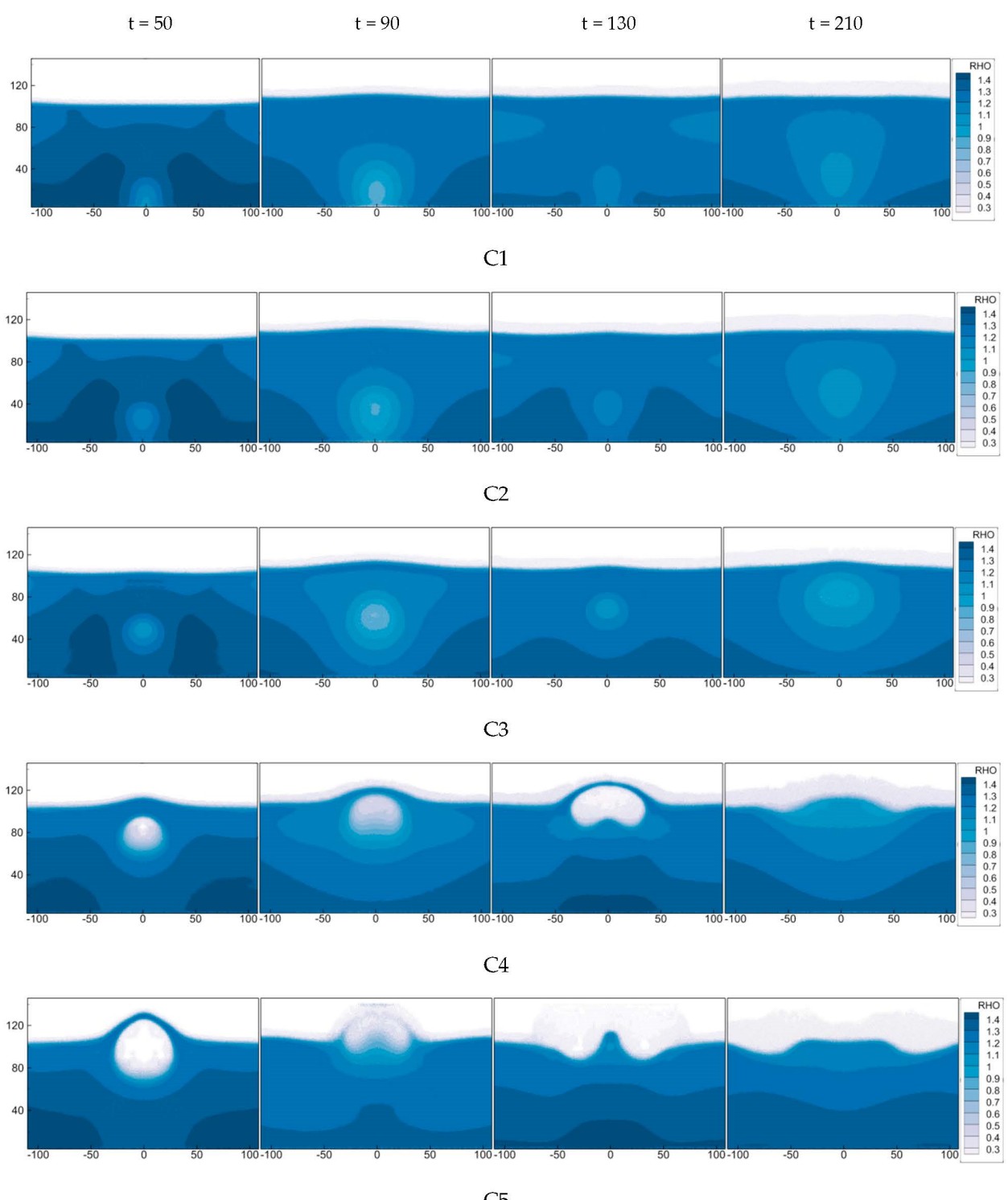

**Figure 13.** The density changes of near-wall bubbles (C1~C3) and liquid-surface bubbles (C4~C5).

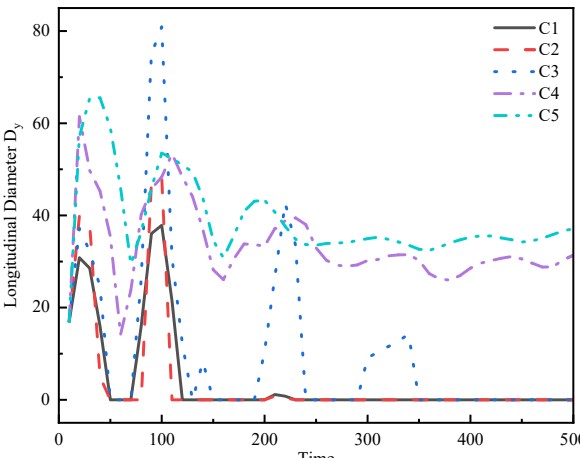

**Figure 14.** The longitudinal diameter changes of near-wall bubbles (C1~C3) and liquid-surface bubbles (C4~C5).

### 5.3.1. Deformation of Near-Wall Bubbles

The deformation of the near-wall bubbles for cases C1~C3 in Figure 13 shows that the vapor bubble increases significantly during the growing period. As it grows, the spherical shape appears to experience a little vertical compression, and it undergoes a period in which it repeatedly grows and collapses. At the beginning, though there are slight differences in the shape of the bubble, the radius of the bubble grows at the same speed. In the process of collapse, the collapse amplitude of the bubbles varies, and the maximum radius of the bubbles is also different, but the duration of the initial bubble is similar.

In Figure 14, for case C3, the bubble has four growth and collapse periods; for cases C1~C2, the bubbles have three growth and collapse cycles. The maximum diameter of bubbles in each period decreases with C1~C3, and the maximum diameter of the second bubbles is larger than that in the other periods.

The bubble symmetry coefficient $\beta$ is introduced to measure the morphological changes of the bubble, where $R_x$ is the transverse radius of the bubble, and $R_y$ is the longitudinal radius of the bubble, as follows:

$$\beta = R_x / R_y \tag{14}$$

Figure 15 illustrates the variation in the symmetric coefficient $\beta$ of the near-wall bubbles for cases C1~C3, and besides the various cycles, the change in $\beta$ is similar. In the pro-phase of the bubble growth, $\beta > 1$, and the bubble is a longitudinal flat oval. In the middle process, as $\beta$ decreases, the deformation of the bubbles undergoes a dramatic change in the collapse period, and $\beta$ increases rapidly after the lowest point. In cases C1~C2, the bubble growth stage is attached to the wall surface; when the vapor bubble is shot out instantaneously, the heat exchange on the solid wall is faster, and when part of it is absorbed, the deformation of the bubble becomes more serious. In case C3, the symmetric coefficient $\beta$ of the non-bonded vapor bubble has little fluctuation scope, and its shape remains spherical.

### 5.3.2. Deformation of Liquid-Surface Bubbles

In Figure 13, in cases C4~C5, the deformation of the liquid-surface bubble appears to extend upwards as soon as it comes into contact with the free liquid surface. When $t = 50$, the gas inside the bubble bends upwards, and the top portion is absorbed by the liquid surface, forming a distinct ellipsoidal tip. Then, the bubble keeps expanding and collapses into a spherical shape. While it is mixed with the liquid surface, a very distinct jet shape is generated at the bottom of the bubble when $t = 120$, as seen in case C5. Eventually, the gas inside the bubble drops quickly due to the gravity force, and then it is pushed back to the surface of the fluid, causing a small oscillation on the free liquid surface.

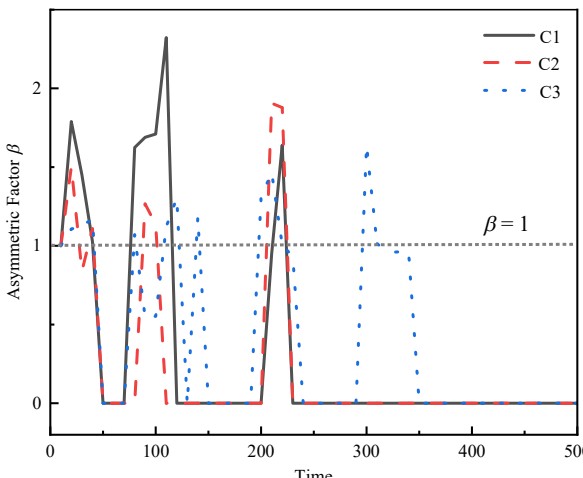

**Figure 15.** The changes of near-wall bubble symmetry coefficient.

Figure 16 shows a graph of the centroid of the Y-axis for the liquid-surface bubbles in cases C4~C5. The results show that after $t = 40$, when the vapor bubble moves nearer to the liquid surface, the higher the gradient of the curve will be. That is, the more rapidly the vapor bubble moves upwards, the more pronounced the jet shape generated by the impact with the free liquid surface. The reason is that as the bubble goes up, it becomes smaller and smaller as it moves closer to the surface. So much of its energy can be transformed into the jet's potential energy and the energy of the bubble's growth and collapse.

### 5.4. Bubble Deformation When Re ≥ 10

On the basis of former research, we analyzed several classical cases of near-wall bubbles and liquid-surface bubbles when Re ≥ 10. Figure 17 shows the change in the bubble deformation and the variation of the flow field density in cases C6~C10.

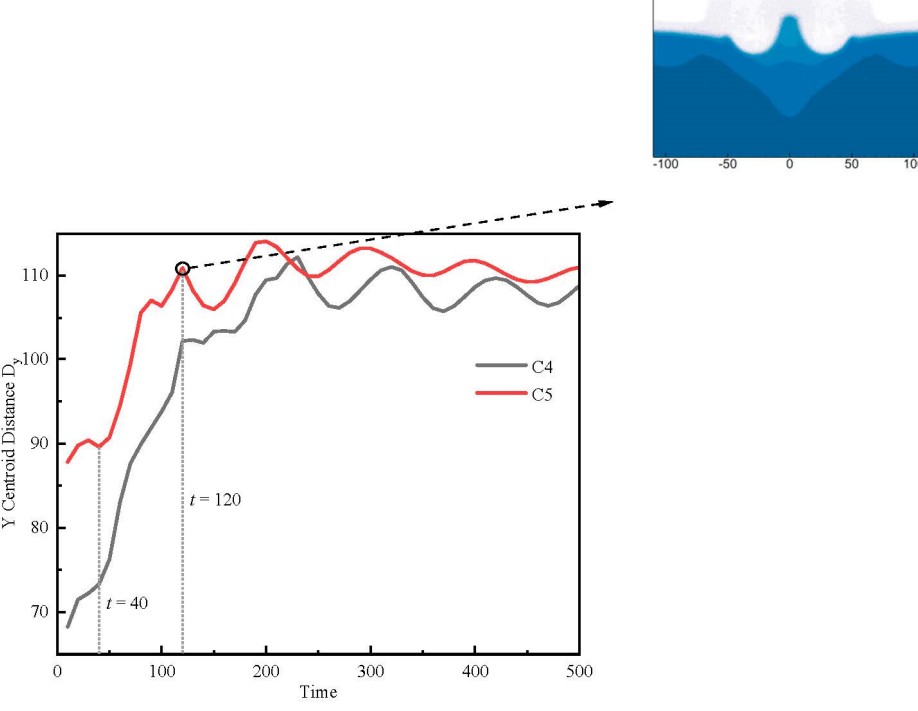

**Figure 16.** The changes of the liquid-surface bubble Y-axis centroid distance.

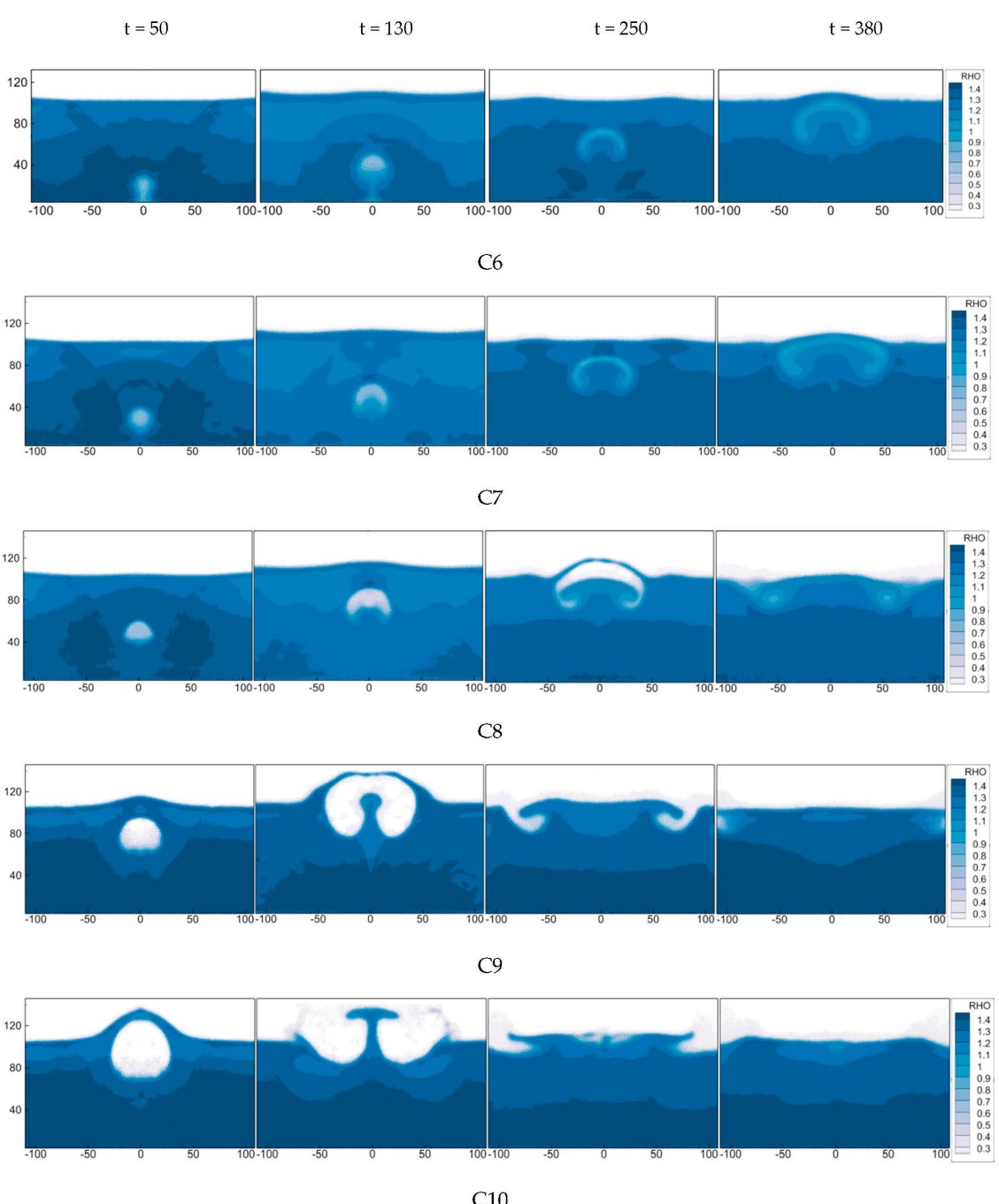

**Figure 17.** The density changes of near-wall bubbles (C6~C8) and liquid-surface bubbles (C9~C10).

The results show that when Re ≥ 10, the shape of the liquid-surface bubbles in cases C9~C10 is similar to that of cases C4~C5 when Re < 10, as seen in Figure 13. Figure 18 compares the changes in the liquid-surface bubble's longitudinal diameter at two conditions. The tendency of variation is quite similar, and there is no obvious difference among them. Therefore, the effect of the Re on the shape of the near-wall bubbles is not apparent. Furthermore, it is found that when Oh < 5 × 10$^{-3}$, at $t = 130$, the jet condition is more evident at case C5 in Figure 13 than at C10 in Figure 17, which is analogous to Sangeeth's experimental conclusion, which states that in the range Oh < 0.02, an increasing viscosity can increase the jet velocity through capillary wave damping [25].

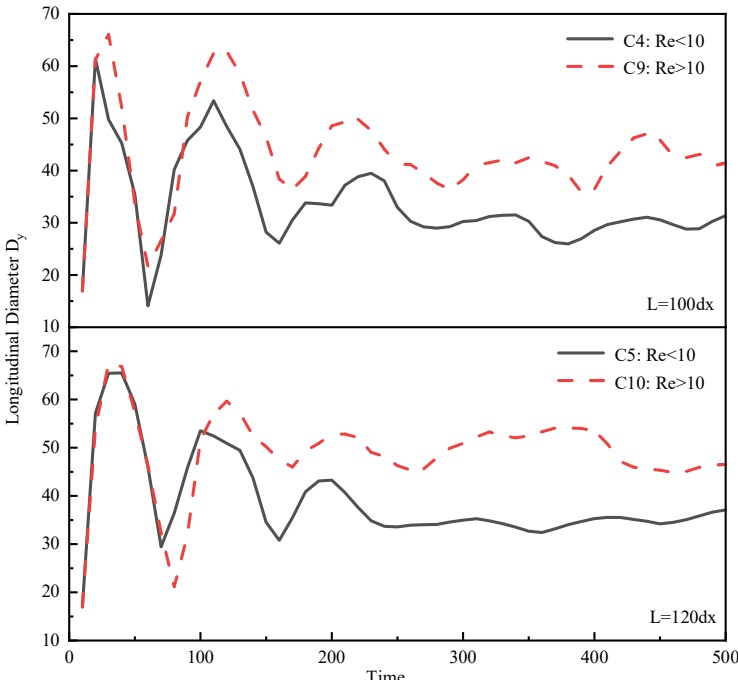

**Figure 18.** Comparison of the longitudinal diameter changes of liquid-surface bubbles with different Re numbers.

For cases C6~C8, we can differentiate the near-wall bubbles into several different stages based on their appearance, which is explained later.

Deformation of Near-Wall Bubbles

In cases C6~C8 of Figure 17, there is a process in the bubble deformation from the initial sphere to the umbrella bubble when *t* = 50 in case C8, and finally, a crack on the free liquid surface.

Figure 19 illustrates the variation of the rate of the near-wall bubbles in cases C6~C8. We find that there is initially a short but fast acceleration, then the bubble oscillates, and the increasing rate decreases as it moves upwards until it hits the free liquid surface. In addition, in cases C6~C8, when the vapor bubble is closer to the free surface, the speed of the bubble is faster, and the bubble oscillation is more intense. This is due to the fact that when the bubble approaches the free surface, there is a smaller amount of gravity potential energy, a larger amount of bubble kinetic energy is required, and a larger flow rate, which results in a more severe oscillation. In the later period, the rate is reduced. This is because the faster the bubble moves to the free surface, the faster it reaches the free surface. After reaching the free surface, the resistance is greater, which causes the increasing rate of the bubble to decrease gradually due to the pressure on the interface liquid and the surface tension of the liquid film. This also has a crucial effect on the subsequent analysis of the characteristics of the bubble deformation.

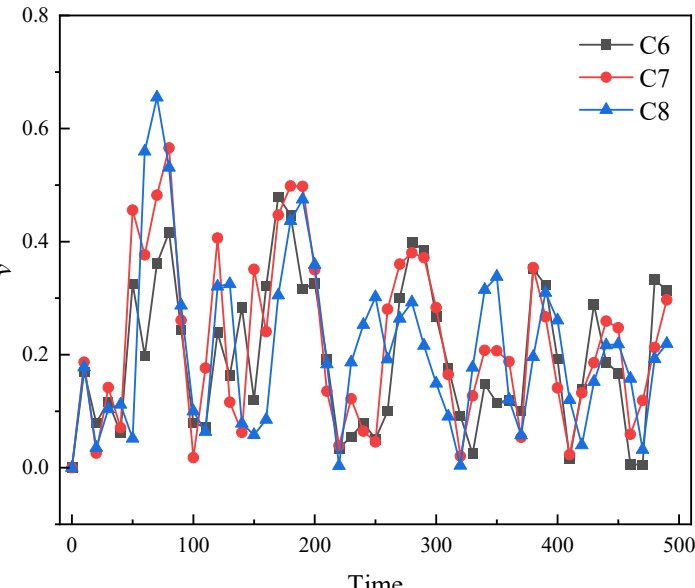

**Figure 19.** The changes of near-wall bubble rising velocity in cases C6~C8.

Based on the morphology characteristics of several stages, the graph of the symmetry coefficient $\beta$ is drawn in Figure 20. Figure 21 shows the stage change chart of case C8, and it is used to classify the stage change of these three classical situations. From Figures 20 and 21, it can be seen that during the initial stage (a), the bubble is subject to extrusion pressure from both the left, right, and the lower sides of the liquid, the bubble changes from an oval to an umbrella shape, and it appears that $\beta > 1$ in the stage (b). Then, as the bubble goes up, the differential pressure in the interior and exterior of the gas–liquid interface at the bottom of the bubble becomes larger. Consequently, the lower part of the bubble quickly contracts inwards and presents a semi-crescent shape as shown in stage (c). Simultaneously, when the bubble approaches the free liquid surface, it pushes it up and forms a liquid film. While the fluid film expands without breaking, a thin "fluid bridge" is formed at the top of the bubble. In the final stage (d), because the surface tension of the "liquid bridge" prevents the bubble from breaking, it merges with the liquid surface.

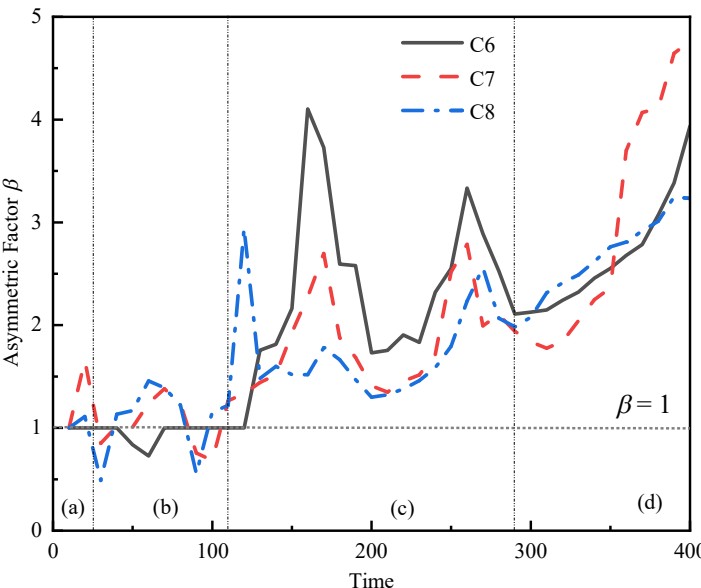

**Figure 20.** The changes of near-wall bubble symmetry coefficient in cases C6~C8.

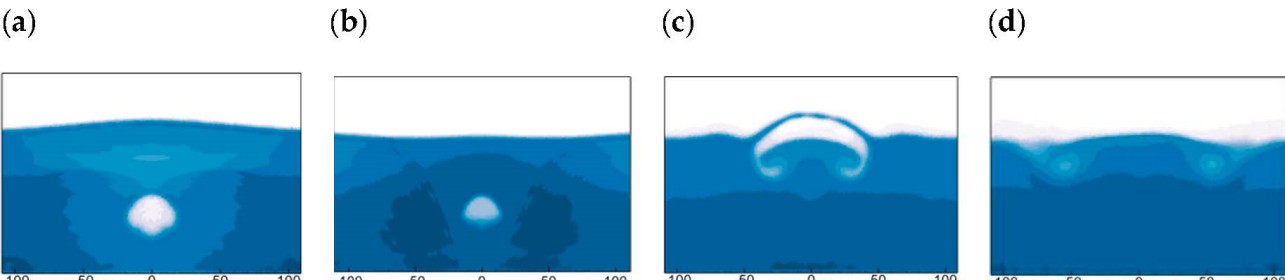

**Figure 21.** The stage change of classical case C8. (**a**) initial stage, (**b**) umbrella shape stage, (**c**) semi-crescent shape stage, (**d**) bubble collapse stage.

## 6. Conclusions

We used the SPH numerical simulation method to directly simulate the deformation and collapse of a vapor bubble near the free surface after being heated and raised from the bottom wall; the effects of the shear viscosity $\eta_s$ and the heating distance $L$ on the growth and collapse processes of the vapor bubble were taken into account. The regularity of the effect of the Re number and the Oh number on the deformation of the vapor bubbles was obtained through a further analysis of several cases, which can be summarized into four major patterns. The classification and mechanism were carried out according to the four major patterns of jet, umbrella, semi-crescent, and spheroid, and the category under each pattern was drawn. The main conclusions are as follows.

For liquid-surface bubbles, the Re number has little influence on them, as there is no significant difference in the specific deformation of the bubbles and the changes in the longitudinal diameter of the bubbles. When $Oh > 5 \times 10^{-3}$, all of them showed a jet shape, and the jet state is more obvious as the shear viscosity increases.

For near-wall bubbles, the Re number has a great influence on the bubble deformation; the shape can be categorized into umbrella, semi-crescent, and spheroid. For $Re > 1.5 \times 10^2$ and $Oh < 3 \times 10^{-4}$, the bubble appears to have an umbrella shape; for $Re < 5 \times 10^0$ and $Oh > 10^{-3}$, the bubble appears to be spheroidal; and for $5 \times 10^0 < Re < 1.5 \times 10^2$, $3 \times 10^{-4} < Oh < 10^{-3}$, the bubble appears to have a semi-crescent shape. Near-wall bubbles experience inhibited longitudinal growth and often collapse at the liquid surface without creating the jet shape. Additionally, the balance of the surface tension and inertia force, influenced by the Re and Oh numbers, contributes to the formation of different bubble shapes.

The spheroidal bubble (cases C1~C3) underwent 3~4 cycles of growth and collapse. As the bubble approached the free surface, its shape became less influenced by the solid wall. The maximum radius of the bubble decreased with each growth and collapse cycle, resulting in less fluctuation in the symmetrical coefficient. Ultimately, the bubble tended to become rounder in shape.

The jet bubble (cases C4~C5) experienced less gravitational potential energy loss as it approached the free surface, resulting in a faster upward movement of the bubble. This, in turn, led to a more prominent jet that occurred upon impact with the free liquid surface.

**Author Contributions:** Conceptualization, Y.C., H.X. and L.Q.; methodology, Q.W. and Y.C.; software, Q.W. and Y.C.; validation, Q.W. and Y.C.; formal analysis, Y.C., H.X. and L.Q.; investigation, Y.C.; resources, Y.C.; data curation, Y.C.; writing—original draft preparation, Y.C.; writing—review and editing, H.X. and L.Q.; visualization, Y.C.; supervision, H.X. and L.Q.; project administration, H.X. and L.Q.; funding acquisition, H.X. and L.Q. All authors have read and agreed to the published version of the manuscript.

**Funding:** This research was funded by the National Natural Science Foundation of China (No. 11972321, No. 91852102).

**Institutional Review Board Statement:** Not applicable.

**Informed Consent Statement:** Not applicable.

**Data Availability Statement:** The data presented in this study are available upon request from the corresponding authors. The data are not publicly available due to privacy reasons.

**Conflicts of Interest:** The authors declare no conflict of interest.

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
