# Peer review of "Vapor Bubble Deformation and Collapse near Free Surface"

_fluids, doi:10.3390/fluids8070187_

Round 1
Reviewer 1 Report
Review of [Fluids] Manuscript ID: fluids-2309358
Vapor Bubble Deformation and Collapse near Free Surface by
Yue Chen, Qichao Wang, Hongbing Xiong and Lijuan Qian
The authors study a bubble arising from thermal heating. The numerical framework looks interesting and could have led to interesting results. However, the presentation is, I’m sorry to say, not very good. Not only is the language incomprehensible in many parts, but the authors seem to have wanted to include as many things as possible. However, this leads to an article which is not very good. I would recommend that the authors seriously reconsider what exactly what they want to show and concentrate on that. Now there are way too many parameters and the article is not of much use to anyone. I do not recommend publication of this article.
The references are a bit of a mess. Sometimes with capital letters, then with full Chinese names or abbreviated. I hope that the research of the authors is not conducted in a similar manner….
Abstact: the sentence “ Results show that Re number has great influence on the bubble deformation at low heating distance: etc” should probably be in the conclusion instead of the abstract.
“Vapor bubble, is a kind of gas which….”, there is a fundamental difference between vapor and gas; a gas satisfies the ideal gas law, the pressure of a vapor only depends on the temperature. Also, this is usually not referred to as ‘cavitation’. Are the authors certain they used the correct jargon?
“the non-dimensional jet speed is represented by Weber Number, the Bond Number and Weber number depend on the dimensionless depth”; this sentence does not read very well. Also the Weber and Bond numbers are not defined.
“By using adjustable low-voltage bubble generator and measuring device,” I know what the authors meant, but this is going to be entirely incomprehensible for the reader. Either delete, or explain better!
“Tang controlled the hydrophobic degree of solid wall by varying the thickness of the gas film,” this sentence is incomprehensible. Is gas = vapour here?
“Navier-Stoke-Korteweg” Stokes?
“The result has certain engineering guiding significance for the current application of vapor bubbles.”; this is very vague. Which advantages are there for which applications?
“represents the surface tension coefficient which is proportional to the gradient energy coefficient”: what does this mean?
Are the simulations done in 2D or 3D?
“Figure 3 is the realization of Zhang's experiment at U = 200V”, I presume this is 200 V? How was the voltage incorporated into the model? What the authors are simulating are essentially explosion bubbles here. It is well known that the vapour pressure inside such a bubble is higher than the one expected from the water room temperature. In that case, the timing would be considerably off. Is this why the time in Fig.2 is given in dimensionless values and in Fig.3 in milliseconds? Moreover, I don’t think Fig.2 and Fig.3 actually resemble that well? The Zhang results show that the bubble rebounds, due to the presence of non-condensable gas. I do not see that in the current simulations? Do the bubbles collapse entirely or what exactly happens? So, the statement “Our results are in good agreement with the experimental results.” is clearly not true?
“This discontinue point is at x = 2.5.” What does this mean?
From Eq.13, the article becomes frankly unreadable….. In Section 4, dimensionless numbers are thrown in without much explanation, using the same symbols as used for the dimensional variables? (rho). Statements like “distance dx = sqrt(?/rho) of the particle is arranged at the bottom of the size of the 400dx × 160dx area” are incomprehensible. How to get back Re, Oh from this mess?
“This is because at lower heating distances, the high temperature in the vapor bubble is absorbed by the solid wall”, but the wall was assumed to be adiabatic?
The results are given in t=100 etc, but the way the equations were non-dimensionalized is not given anywhere. Or is it t=100 seconds? How is G=-0.02 to be interpreted? Or “the heating radius is r = 12dx”?
The simulations were performed with a free surface and a solid surface at the bottom. Furthermore gravity, viscosity, Re, Oh and ?? were changed. This leads to too many parameters in the model. The reader will frankly get lost.
Fig.6 and corresponding text should be in an appendix.
There are not enough frames in Fig.2 to see what is happening. Same comments for Fig.9. In Fig.8 upper left image, there is not even a bubble?
Fig.9: Does the bubble vent to the free surface, or does it collapse under the surface?
Fig.15: How to interpret a longitudinal radius of 60 to 80? What does this mean?
“the bubble is semi-crescent, which is consistent with the shape of the bubble in the literature [30].” Ref.30 is Bhaga and Weber which studied a totally different physical system. The current bubble dynamics are totally different!
The English language is quite bad in many places and should be fixed up. Please employ someone to fix it up.
Reviewer 2 Report
The authors used the Smooth Particle Hydrodynamics (SPH) method to simulate the bubble behaviors near a free surface. After verification with a lased-induced bubble interaction with a free surface, the authors report four interesting deformations of the bubble for different Reynolds number and Ohnesorge number.
The validation of the numerical methods in the manuscript is not adequate and thus the results are not convincing for the present. The comments/questions of the reviewer are as follows:
Major points:
1. To the Referee’s understanding, the experimental example by Zhang et al. (spark-induced, not laser-induced) does not have a bottom rigid wall as near as illustrated in Figure 1. Moreover, the adiabatic boundary condition is not fully explained in the main text.
2. In lines 175-182, all the parameters seem to be dimensionless, however without adequate explanations and definitions.
3. The validation between experiments and simulations is recommended to be arranged in the same figure and in the same timescales. Could the authors combine figures 2 and 3 and compare the frames at the same time instants?
4. In Figure 4, it also seems not adequate to compare the dimensionless displacements of the interface for validation. Otherwise, to make the validation more convincing, the evolution of the bubble radius with time should be provided.
5. The contents in Figure 1 and Figure 7 have many similarities. Could the authors combine the figures into one?
6. The different shapes of the bubble deformation are interesting, however, not convincing enough. Are there any published papers especially on experiments stating the bubble deformation? Is it possible to control Re and Oh in experiments?
Minor points:
1. One point on citation format in the main text. The numbers are recommended to follow the names, e.g., lines 45-46 “Gonzalez-A et al. [10] studied …”. The same is for the rest.
2. The parameters r and \alpha_{dim} in Eq.(7) may need to be defined at their first appearance.
3. Line 168, the authors may need to explain further what U=200V means.
4. What does the excess heat mean in line 180 and why is it equal to 12?
5. Is the gravity G equal to 0.02 (line 183) or -0.02 (line 236)?
6. The line numbers and the tables overlap each other. The authors may have a check.
Reviewer 3 Report
This is a computational fluid dynamics study, where the authors study the bubble dynamics by means of the so-called Smooth Particle Hydrodynamics method.
The authors identify the Re and Oh numbers as the control parameters in their system, and are able to classify the types of geometries that cavitating bubbles can develop under the conditions they study. The compare with experimental data, with which they find at qualitative agreement.
I say at least because there is currently no way to assess the degree of quantitative agreement since the authors present their numerical data sets in reduced magnitudes (whose units the do not specify), whereas the experimental data from bibliography that they show are in dimensional form.
Additionally, English needs revision in several parts of the manuscript. Please revise this point.
Most importantly, their Smooth Particle Hydrodynamics method is not explained at all in their manuscript. This is very apparent since understanding the features of this numerical method, in my opinion, is key for this work.
Therefore, before I can accept the manuscript for publication, this issue must urgently be addressed.
Other comments:
* Line 15: "After the verification" -> "After verification".
* Line 25. Please revise English here and elsewhere.
* Line 62. Please explain in detail the SPH method.
* Line 138: Full stop after equation. Provide reference and justification on why this is so.
* Line 167: "with free surface in Zhang et al. [17], and compare with the experimental phenomena" -> "with free surface and compare with experimental [17]".
* Line 175 and henceforth: Please specify simulation units.
* Figure 2: for better represetantion and meaningful comparison, please use same units in Figures 2 and 3.
* Line 209. The two first sentences are actually not necessary.
*Line 214. I get lost in the transition from end of previous page. "discontinue point"? have the authors mentioned what that is about before? dont think so.
Round 2
Reviewer 1 Report
I repeat what I said before and still think this work is acceptable, but if the editor thinks it should be published, then I'm fine with that. "I would recommend that the authors seriously reconsider what exactly what they want to show and concentrate on that. Now there are way too many parameters and the article is not of much use to anyone. I do not recommend publication of this article." Moreover, it now turns out that the simulations are in 2D, while the real world phenomena that are being described are in 3D.....
Author Response
Thank you for your attention and valuable suggestions for our paper. Please see the attachment for the detailed response.

Reviewer 2 Report
The authors answered most of the questions raised by the reviewer. However, the reviewer suggests that they address the following points before the manuscript can be considered to be published.
English writing needs improvement. For example, the font of symbols in the equation and the main text should be kept the same, and following the convention.
Author Response

(The authors gave the same response as above.)

Reviewer 3 Report
The authors have addressed my comments satisfactorily and I can tell they made a significant effort in order to improve the quality of English.
While till some improvements can be made, as the other Referees pointed out, my understanding is that these issues do not affect the essentials of the results presented here.
My advice is that they undertake this task of taking into consderation those suggestions for improvement. After that, I have no doubt that this paper can be published in fhe Fluids journal.
For this reason, I am ready to accept it for publication.
Author Response

(The authors gave the same response as above.)
